# Benefit Analysis of Gamified Augmented Reality Navigation System

**Chun-I Lee**

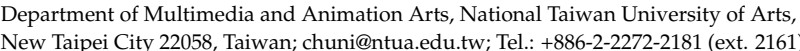

Department of Multimedia and Animation Arts, National Taiwan University of Arts,
New Taipei City 22058, Taiwan; chuni@ntua.edu.tw; Tel.: +886-2-2272-2181 (ext. 2161)

**Abstract:** Augmented reality (AR) technology has received much attention in recent years. Users can view AR content through mobile devices, meaning that AR tee no significant differences between the exhibition-viewing behaviors associated with the game mode and the frchnology can be easily incorporated into applications in various fields. The auxiliary role of AR in exhibitions and its influence and benefits with regard to the behavior of exhibition visitors are worth investigating. To understand whether gamified AR and general AR navigation systems impact visitor behavior, we designed an AR app for an art exhibition. Using GPS and AR technology, the app presents a virtual guide that provides friendly guide services and helps visitors navigate outdoor paths and alleys in the exhibition venue, showing them the best route. The AR navigation system has three primary functions: (1) it allows visitors to scan the exhibit labels to access detailed text and audio introductions, (2) it offers a game mode and a free mode for navigation, and (3) it collects data on the exhibition-viewing behavior associated with each mode and uploads them to the cloud for analysis. The results of the experiment revealed no significant differences between the exhibition-viewing time or distance travelled in the two modes. However, the paths resulting from the game mode were more regular, which means that the participants were more likely to view the exhibition as instructed with the aid of gamified AR. This insight is useful for the control of crowd flow.

**Keywords:** augmented reality; gamification; navigation system; exhibition-viewing behavior

## 1. Introduction

This study is an extension of a navAR system developed by Lee, Xiao, and Hsu [1]. It enables the design of AR guide apps from the perspectives of visitors, content providers, and researchers. The system records user behavior and then automatically uploads it to the cloud for analysis. The system has been applied to a library AR book-searching experiment, the results of which revealed that with the aid of AR, the book-searching paths of users are more directed [2]. To understand whether gamified processes possess potential guidance functions for AR guide systems, the researchers developed an AR navigation app (Appendix A) for the Greater Taipei Biennial of Contemporary Art 2020 exhibition, "Authentic World", which was held by the Yo-Chang Art Museum in 2020. The researchers then observed the exhibition-viewing behaviors of users under the influence of gamified AR navigation. A total of 20 artworks were displayed at "Authentic World" (Appendix B), and they were placed in different independent spaces scattered throughout NTUA Art Village and 9 Art Space, which are large outdoor exhibition spaces.

To prevent AR from interfering with the exhibits and because visitors needed to remain moving and outdoors, we created a virtual AR guide based on the theme and style of the exhibition that users could follow along exhibition paths. The AR image positioning of this guide was achieved using ARCore and ARKit in conjunction with Mapbox SDK for outdoor GPS positioning. The location information on a two-dimensional (2D) map and the corresponding coordinates of the virtual AR guide in the physical venue (Figure 1) were employed to enhance the accuracy of navigation.

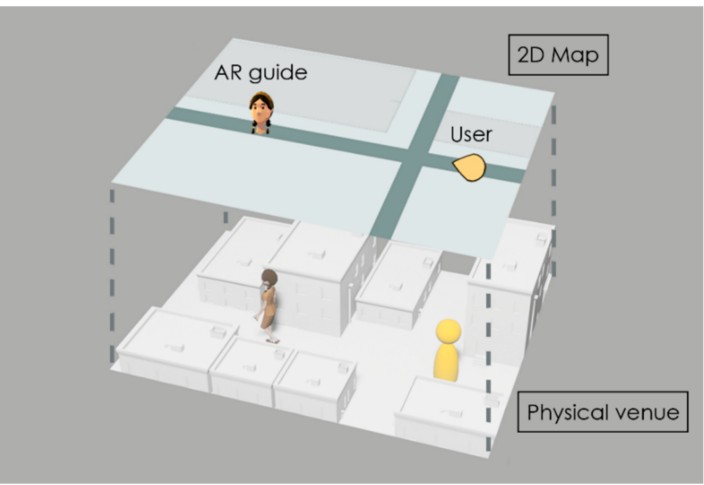

**Figure 1.** Schematic diagram of cyber-physical integration.

## 2. Related Work

### 2.1. Benefits of AR

The relevant literature has shown that AR can generate interest in visitors in virtual and real learning environments [3]. However, despite its benefits, AR can cause users to become too absorbed in its effects to pay attention to the physical environment. Therefore, developers must avoid making the AR content the focus of the experience [4] and augment the present reality of the user rather than remove them from the scenario [5]. Swan and Gabbard [6] discovered that out of 266 papers on human–machine interfaces, 38 studies (14.3%) focused on user-centered AR designs, whereas only 21 studies (7.9%) were aimed at experiments with general users using AR. Dünser et al. [7] also pointed out that only 10% of 161 studies involved AR experiments. Scant research on AR user experiences makes it difficult to determine the benefits of AR. A study pointed out that 291 influential AR user studies from 2005 to 2014 found that hand-held AR user studies have gradually increased, 76% of them were mainly conducted in laboratory experiments, and 15% of them were conducted in a natural environment or field study [8]. This also shows that user research related to AR technology in outdoor applications is still quite scarce. Existing analyses of AR user benefits have been applied the questionnaire survey method, the observation method, semi-structured interviews, and video analysis [5,9]. However, the questionnaire survey method, the observation method, and semi-structured interviews all collect the perspectives of users, making it difficult to objectively analyze user behavior. Video analysis allows researchers to obtain data on user paths and tracks through large numbers of videos [10,11], and additional sensors placed in the pockets [12] or shoes [13] of participants. However, this requires additional equipment, which is inconvenient for research analysis. In recent years, AR technology has been used not only as an aid but also a research analysis tool. Lee et al. [1] developed the AR behavior analysis system navAR to understand the differences between the AR-aided book-searching behavior and the general book-searching behavior of library users [2].

### 2.2. AR Navigation Technology

The most common application of AR to exhibitions is granting new life to static displays. For instance, artist Alex Mayhew created the ReBlink [14] app for the Art Gallery of Ontario in Toronto, Canada, and designed an AR animation for each well-known painting in the exhibition hall. Using image recognition technology and the cameras of tablet computers, the animations could be played over the actual painting, adding an element of fun to the exhibition. The Detroit Institute of Arts incorporated Google Tango technology [15] to enable users to see the bones inside of mummies by scanning with their smartphones, and the cyber-physical integration was relatively stable. However, Tango



technology requires that the mobile device be equipped with a compatible depth-sensing lens. Thus, it cannot be used with general mobile devices, which made it difficult to promote. Not long after, it was replaced by Google ARCore and Apple ARkit, which perform AR positioning using general smartphone camera lenses [16]. The "Yōga: Modern Western Paintings of Japan" exhibition hosted by the Museum of National Taipei University of Education [17] incorporated the ARKit technology developed by Apple into their exhibition app to incorporate AR into the representative works of the exhibition experience. AR technology was the focus of this event; instead of the more common image scanning method, there were only frames in the physical venue, and all of the works were viewed on smartphones via SLAM positioning.

### 2.3. Gamified AR Navigation

Gamification is the incorporation of game design elements and mechanisms into non-game fields [18]. The gamification process can be used to solve problems [19] and to improve user actions [20] and learning abilities [21]. The problem faced in general exhibitions is ensuring exhibits appeal to visitors, which influences their willingness to approach them and can affect the evenness of people flows at an exhibition. It is thus hoped that gamification can motivate visitors to view exhibits. Gamification does not replace the gamified target. Hamari and Koivisto [22] employed the Dispositional Flow Scale–2 (DFS-2) to determine the significance of gamification to the flow in autotelic personality traits. The results indicated that it was significant to autotelic experience, clear goals, immediate feedback, and control and the challenge-skill balance but less significant to time transformation, merging action-awareness, and loss of self-consciousness. This shows that when gamifying navigation, we must remember that the purpose is not to immerse the users in the gamified content but to give users clear goals and actions, intuition, and feedback so that it promotes the goal rather than steals the spotlight.

AR technology can expand the imagination of users, and adding gamification can stimulate them to complete more tasks [23]. Ioannis Paliokas et al. [24] used AR application to enhance the viewing experience at the Silversmithing Museum by adding gaming and educational elements, and the evaluation found that this approach helped improve museum viewing experience satisfaction and learning outcomes. Philipp [25] used three 3D virtual guides to guide visitors through the Museum of Celtic Heritage's collection. Göbel and Sauer [26] used mobile devices and AR gamification to recreate ancient creatures. Hammady et al. [27] applied AR technology to create an Egyptian mythology game for the Egyptian Museum. There are also educators and developers using mobile AR to create educational games, allowing secondary school students to learn the history of archaeological sites through a guided tour [28]. AR technology is not only used for indoor exhibition halls in museums, but also for outdoor tours. For example, Razvan Gabriel et al. [29] used mobile augmented reality to design a MAR application in three different outdoor locations in Europe, using 3D models to presenting information on the life and history of the Roman poet Ovid. In addition to combining AR images and gamification, the Blast Theory [30] team in the UK created "Ghostwriter", a gamified navigation work with voice narrations as the augmenting element. This work involves a pre-recorded voice telling a story set in the museum. When visitors call a certain number, a woman's voice coming from the other end of the line directs visitors to various exhibits, explains her relationship with the exhibit, and connects the exhibits together. This is a good example of incorporating the navigation process into the gamified AR content. However, investigating the effectiveness of the navigation and improving it as necessary requires systematic recording and analysis of user behavior.

## 3. Methodology

To understand the benefits of incorporating AR into exhibition navigation, we developed an AR navigation system app to record and analyze the exhibition-viewing behaviors of the users. Our methodology can be divided into two aspects: the AR navigation system and the experimental design.

### 3.1. AR Navigation System

To avoid creating an experimental process that would ruin the exhibition-viewing experiences of users, we tailored the navigation app to the "Authentic World" exhibition. The primary functions of the map included artwork label scanning, two navigation modes, and the uploading of behavioral data to the cloud. The details of the functions are as follows:

### 3.1.1. Artwork Label Scanning

The art pieces were displayed at different exhibition buildings. Visitors could go to the different exhibition buildings (Figure 2a) and use the AR camera lens to directly scan the label on a wall next to any artwork to obtain a detailed text introduction on their smartphone screen and listen to an audio introduction (Figure 2b). Each time they scanned an artwork, it was marked in a different color on the 2D map on their phone to indicate it had been viewed. The labels were also printed on paper handouts, enabling visitors to collect the introductions as souvenirs. Thus, even after the exhibition ended, it was possible for the visitors to scan the labels and access the artworks.

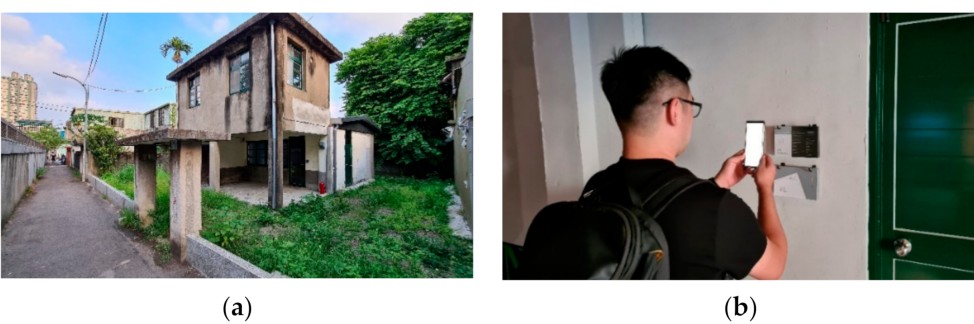

(**a**)                                                                 (**b**)

**Figure 2.** (**a**) 9 Art Space exhibition building; (**b**) AR scanning of artwork label.

### 3.1.2. Two Navigation Modes

There were two types of navigation modes: a game mode and a free mode. Users were free to choose either of the two modes to guide them. In the game mode, users could switch to AR mode (Figure 3a) or 2D map mode (Figure 3b) at any time and follow an AR guide through the alleys in the exhibition venue (Figure 3c). At each exhibition building, they could scan the label of each artwork using the camera lens, collect all 20 artworks, and take a photo with the virtual AR guide. Using the photo, they could obtain an AR souvenir sticker at the information desk and take the virtual guide home as a souvenir and reward.

The free mode provided a general GPS location search and navigation function for each artwork. Visitors could decide which artwork they wanted to see (Figure 4a), and the system would show the shortest route to it (Figure 4b). Aside from providing introductions to the artworks, scanning the labels offered no rewards.

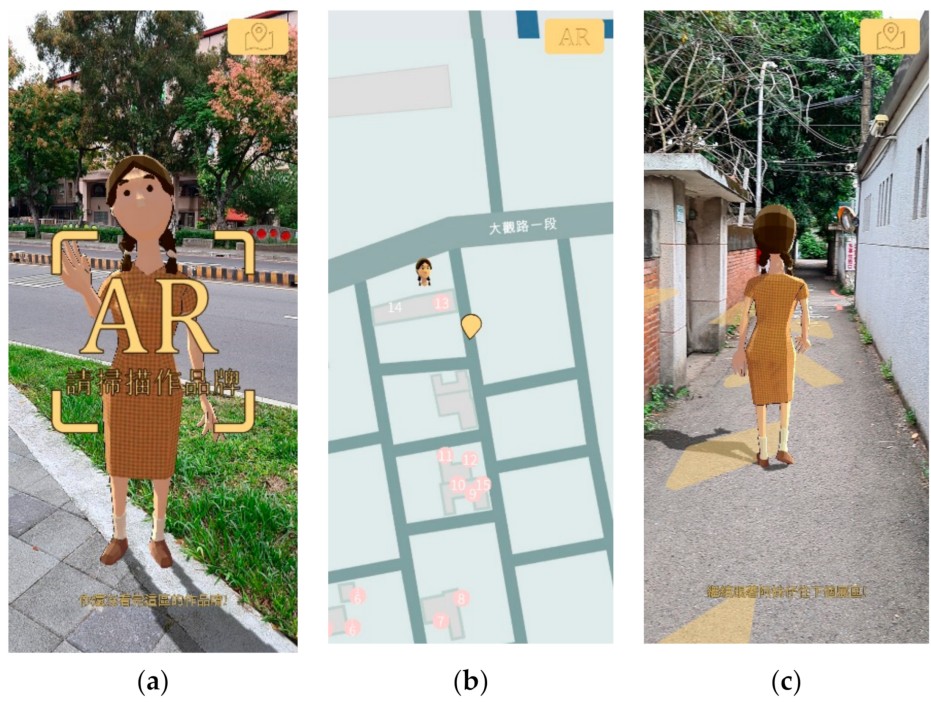

|  |  |  |
|:--:|:--:|:--:|
| (**a**) | (**b**) | (**c**) |

**Figure 3.** (**a**) AR mode (The on-screen prompts the visitors to scan the label of artwork); (**b**) 2D map mode; (**c**) AR guide walking through an alley (The on-screen prompts the visitors to follow the AR guide).

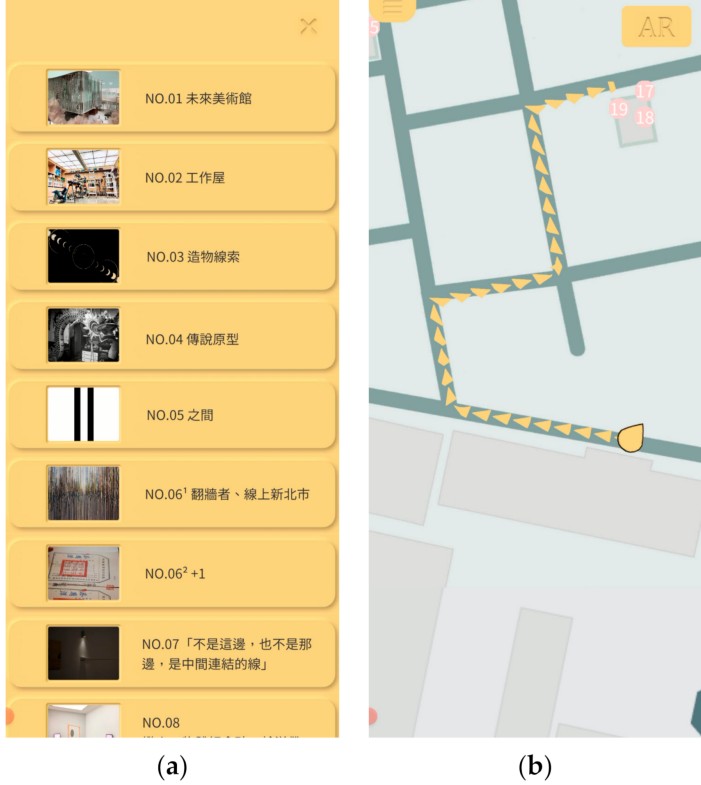

|  |  |
|:--:|:--:|
| (**a**) | (**b**) |

**Figure 4.** (**a**) List of artworks; (**b**) shortest-route navigation.

### 3.1.3. Uploading of Behavioral Data to the Cloud

Contingent upon the consent of the participants, the time and distance travelled by each were uploaded to a prepared Google Sheets form. The system automatically generated the path of each visitor and uploaded it to Google Drive for subsequent analysis.

### 3.2. *Experimental Design*

For the convenient and realistic testing of the effectiveness of AR on navigation, we conducted our experiment at NTUA Art Village and 9 Art Space, which are areas belonging to the Yo-Chang Art Museum near our campus at the National Taiwan University of Arts. The exhibition displayed a total of 20 artworks, and the suggested path was as shown in Figure 5. Since the art village where this exhibition is located is an art village transformed from a declining village, there are still many houses that have not been used and sorted out. Therefore, this exhibition does not include all areas on the map. The locations shown in Figure 5 are the areas where visitors are expected to go to, and the rest of the areas are not the locations we want visitors to reach. We adopted a between-subject design for the experiment, allowing the visitors to choose between the game mode and the free mode to view the exhibition. Figure 6 presents a flow chart of the experiment design. As our experiment had to coordinate with the exhibition period, all of the participants were general members of the public who had come to see the exhibition. Thus, convenience sampling was employed. The participants downloaded the navigation app by scanning the guidebook, promotional materials at the exhibition, or the QR code on the exhibition website and then followed system instructions to use the app. No staff members assisted them during their visit. To collect the exhibition-viewing behavior data of the visitors, the app asked for permission to record relevant information during the visit and upload the visitor's path, distance travelled, and time data up to a cloud for subsequent viewer behavior analysis. The app made it clear that the purpose of this was to understand the behavioral differences resulting from the two modes so as to improve future exhibition experiences. In coordination with the exhibition, our study period ran from 30 October 2020 to 31 December 2020. A total of 99 visitors participated in the experiment, with 45 participants using the free mode and 54 participants using the game mode.

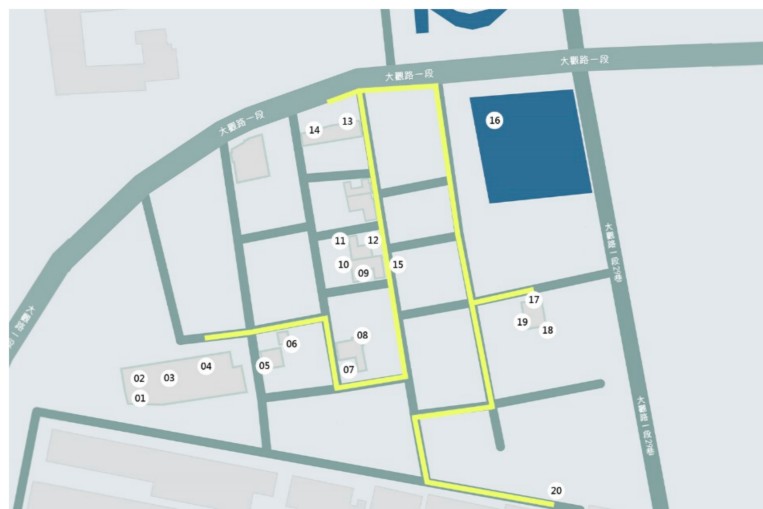

**Figure 5.** Locations of artworks and suggested route (The numbers in the figure are the artwork numbers).

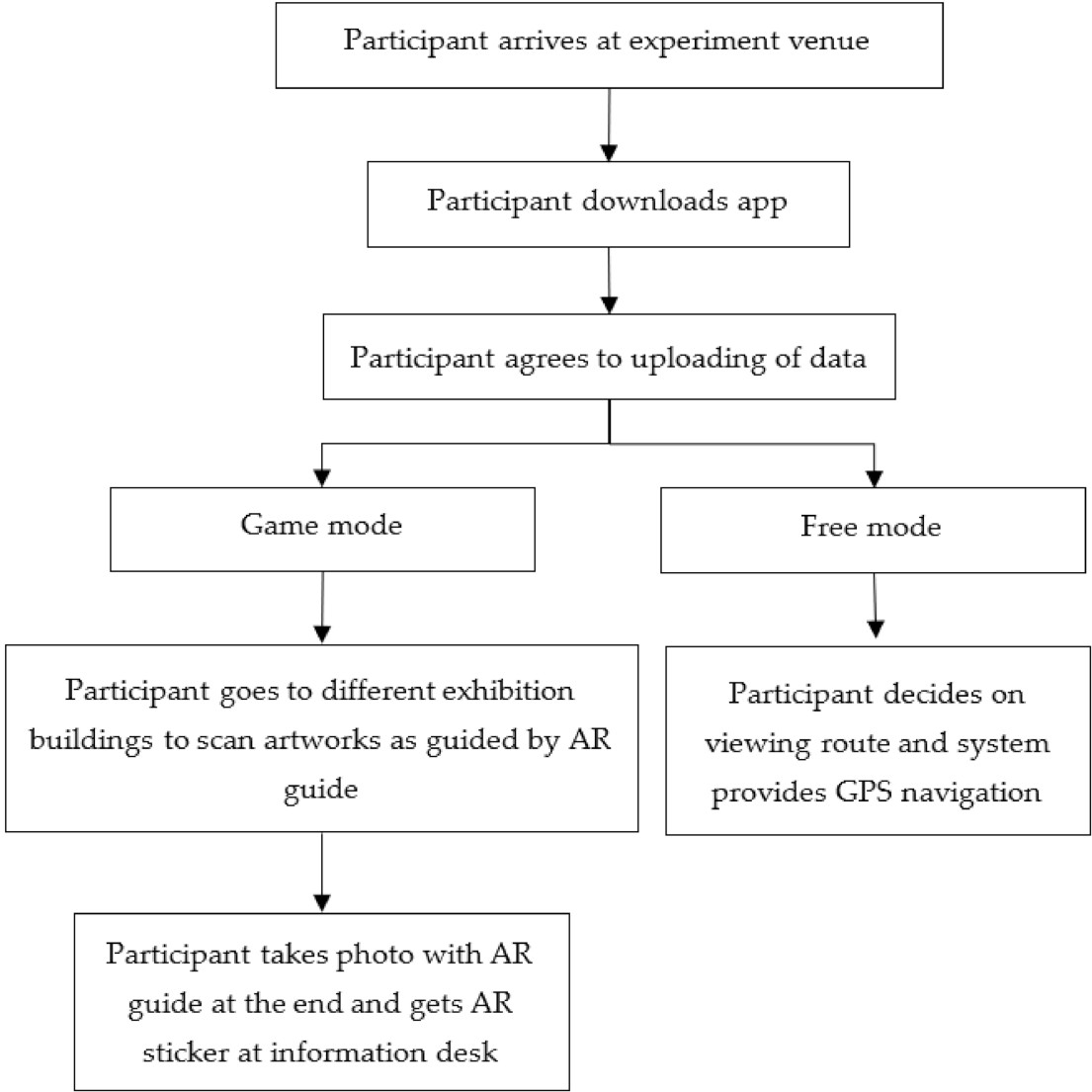

**Figure 6.** Flow chart of experiment design.

## 4. Data Analysis

To understand the differences between users using the game mode and the free mode in terms of exhibition-viewing behavior, we analyzed the collected data using independent *t*-tests. The time and distance travelled by the participants were imported into SPSS for analysis. Our hypotheses and results were as follows:

**Hypotheses 1 (H1).** *There were no significant differences between the exhibition-viewing behaviors associated with the game mode and the free mode.*

**Hypotheses 2 (H2).** *There were significant differences between the exhibition-viewing behaviors associated with the game mode and the free mode.*

### 4.1. Viewing Time

Analysis of the time that the participants spent at the exhibition (Table 1) revealed a test statistic of $t(97) = 0.962$ ($p > 0.05$). Thus, the null hypothesis (H1) could not be rejected. At an $\alpha = 0.05$ level of significance, the exhibition-viewing time resulting from the game mode and the free mode presented no significant differences.

**Table 1.** Summary table of *t*-test of exhibition-viewing time.

| Mode | No. of Samples | Time (s) | | | | |
|------|----------------|----------|---|---|---|---|
| | | Mean (n) | Standard Deviation (SD) | *t*-Value (t) | Degree of Freedom (df) | *p*-Value (p) |
| Free | 45 | 340.20 | 606.334 | 0.962 | 97 | 0.339 |
| Game | 54 | 243.78 | 382.659 | | | |

*4.2. Distance Travelled*

Analysis of the distances travelled by the participants at the exhibition (Table 2) revealed a test statistic of t(59.099) = 1.451 (*p* > 0.05). Thus, the null hypothesis (H1) could not be rejected. At an α = 0.05 level of significance, the exhibition-viewing distance travelled resulting from the game mode and the free mode presented no significant differences.

**Table 2.** Summary table of *t*-test of exhibition-viewing distance travelled.

| Mode | No. of Samples | Distance Travelled (m) | | | | |
|------|----------------|------------------------|---|---|---|---|
| | | Mean (n) | Standard Deviation (SD) | *t*-Value (t) | Degree of Freedom (df) | *p*-Value (p) |
| Free | 45 | 329.889 | 549.8836 | 1.451 | 59.099 | 0.152 |
| Game | 54 | 201.050 | 250.7699 | | | |

*4.3. Path Analysis*

On the whole, analysis of the path data automatically uploaded by the smartphones of the participants revealed that the initial starting points of the participants were scattered, regardless of navigation mode. However, as shown in Figure 7, where each color indicates the path data of a single participant, the paths of the participants who used the game mode (Figure 7a) were more regular and consistent than those of the participants who used the free mode (Figure 7b), thereby showing that gamified navigation had a greater guiding effect.

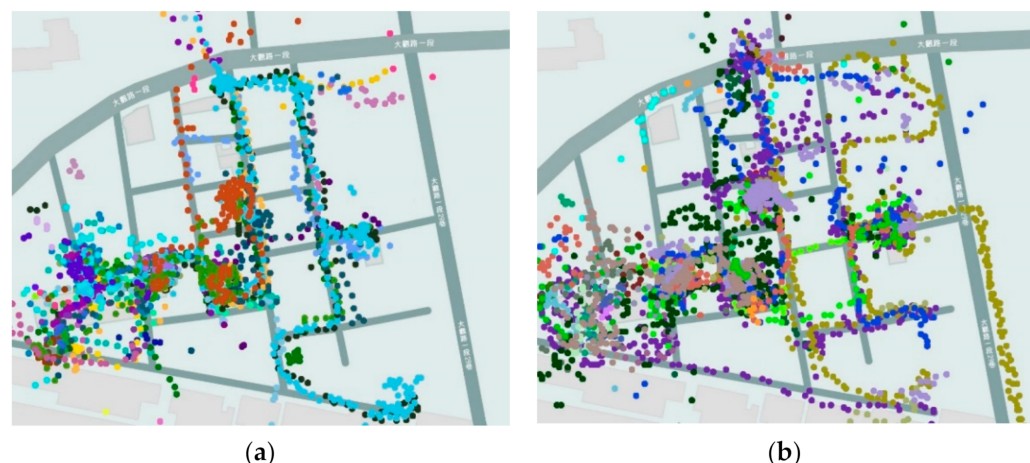

(**a**)                                                    (**b**)

**Figure 7.** (**a**) Path distributions of participants using game mode (N = 54); (**b**) path distributions of participants using free mode (N = 45).

## 5. Discussion

*5.1. Analysis Results*

- Due to current issues with technologies combining AR and GPS, the AR guide's position shifted even when the user was standing still. Thus, this method is not recommended for outdoor venues. These contexts would be better served by fixed-location display methods, in which the virtual guide is displayed after images are scanned at each exhibit.

- Analysis showed no significant differences between the participants in terms of time or distance travelled, meaning that gamification did not impact the personal pace of the participants or the time they spent viewing the exhibition, and the overall distance that the participants travelled did not differ significantly. This means that using gamification to control the time spent viewing exhibits or distance travelled by visitors would be difficult.
- The paths presented by the participants using gamified AR were more regular, and therefore we can speculate that visitors follow the gamified path instructions while viewing exhibitions. This is a significant finding, as it can be applied to divert crowd flows. However, the starting points of the participants remained scattered. We speculate that this was because the participants needed time to familiarize themselves with the map and because their smartphone hardware and GPS positioning conditions can affect AR stability. This requires improvement.

*5.2. Research Limitations*

- Since this study is still based on mobile devices that are easily available to users and support general AR technology, devices that can enhance AR positioning quality or experience such as Bluetooth beacons, smart glasses, or edge-based SLAM applications are not within the scope of this study.
- The objective of this study was to represent the real behavior of exhibition visitors using a gamified AR navigation system. Thus, no guidance or assistance from staff members was provided, which means interruptions or other behaviors beyond exhibition viewing were unavoidable. These behaviors were not within the scope of this study.
- Exhibitions manifest in a myriad of forms. This study focused on AR-aided navigation in which visitors moved from exhibit to exhibit using a gamified approach and by following the guidance of a virtual AR guide outdoors. The results provide reference for other exhibitions and studies employing a similar approach but cannot be generalized to all types of AR navigation.
- The limitations of existing AR and GPS technologies mean that not all of the participants had optimal AR guide quality and effects. These limitations included varying smartphone hardware specifications and external environmental factors such as internet connection quality, weather, and overhead cover. These may affect the persistence and promotion of AR navigation system use.

## 6. Conclusions

Without the installation of additional spatial positioning equipment, we developed a behavior analysis system for the smartphones of exhibition visitors to analyze their exhibition-viewing behavior while using an AR navigation system. This enabled the collection of real-world data beyond a laboratory-like environment. We analyzed and compared the behaviors resulting from two different navigation modes to determine their differences and benefits to serve as reference for navigation system improvements. The analysis revealed no significant differences in exhibition-viewing time or distance travelled between the general and gamified modes. This means that using gamification to control the time spent viewing the exhibits or distance travelled by visitors would be difficult. However, the paths presented by the participants using gamified AR were more concentrated, which means that these participants could more easily find their way under the AR guidance. On this basis, we infer that gamified AR has guiding effects and can successfully divert crowd flows. Limitations in pairing existing AT technologies with the hardware of mobile devices prevented optimal navigation quality; however, it is foreseeable that as AR and GPS combinations and mobile device technology mature or when lightweight AR glasses become common, AR and gamified navigation systems will be able to offer richer, more immersive exhibition-viewing experiences.

**Funding:** This research was funded by the Ministry of Science and Technology (MOST), grant number MOST 109-2410-H-144-003-MY2.

**Institutional Review Board Statement:** All subjects gave their informed consent for inclusion before they participated in the study. The study was conducted in accordance with the Declaration of Helsinki, and the protocol was approved by The Research Ethics Committee of National Taiwan University (protocol code: 202005ES083 and date of approval: 21 August 2020).

**Informed Consent Statement:** Informed consent was obtained from all subjects involved in the study.

**Data Availability Statement:** Publicly available datasets were analyzed in this study. This data can be found here: https://drive.google.com/drive/folders/1KbAF_kfeD4ApJbMxpZLxXWEnhsgsROeR?usp=sharing (accessed on 1 February 2022).

**Acknowledgments:** This research was supported in part by the Ministry of Science and Technology of Taiwan, National Taiwan University of Arts, Yo-Chang Art Museum, Innovation Center for Art and Technology, and NTUA Experimental Game Lab.

**Conflicts of Interest:** The author declares no conflict of interest. The funders had no role in the design of the study; in the collection, analyses, or interpretation of data; in the writing of the manuscript; or in the decision to publish the results.

## Appendix A

Video of the process of AR navigation: https://youtu.be/fHPdRrYlP1E (accessed on 1 February 2022).

## Appendix B

Official website of the Authentic World exhibition: https://www.gtbca.com/home (accessed on 1 February 2022).

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
