# Peer review of "Benefit Analysis of Gamified Augmented Reality Navigation System"

_applsci, doi:10.3390/app12062969_

Round 1

Reviewer 1 Report

This paper presents an extension of an existing system of AR navigation. A virtual AR guide has developed, which can be followed by visitors of an exhibition. One important aspect of the work is to prevent AR to interfere with the exploration of physical space. 

Users behaviour was recorded to compare the virtual guide navigation and the free navigation mode. Results showed that GPS is not precise enough to ensure robust registration of the virtual guide. Maybe technologies as Bluetooth beacons should be considered?

The description of the population participating to the study is missing and the analysis of results is weak. It seems that the trajectoires are only analysed visually. In that case, if we trust figure 7, we can say that the "gamification" have drawbacks, because trajectoires in game mode don't cover the whole exhibition space. Maybe users only visit the checkpoints to have the reward?

Bibliography must be corrected: there is a lack of information, formatting is not uniform, references 14 and 17 are truncated, last reference seems strange. In addition, there is no recent references, the most recent is from 2017. 

Reviewer 2 Report

The matter of the proposed paper is important and interesting. The approach to verify an influence of gamification on human behavior is in general valid. However, the major flaw of the presented work is relating to quite old publications. Both AR and gamification topics are quite hot nowadays, so basing on 10-years-old papers is not acceptable. The only two current papers (published in 2019 and 2020) do not cover the most important issues in the paper. It should be noticed that there are other papers, recently published, and covering very similar topics (e.g.  A Gamified Augmented Reality Application for Digital Heritage and Tourism, Paliokas et al, 2020). Moreover, the Authors refer to relatively old technologies, ignoring more recent solutions like smartglasses or edge-based SLAM applications. These technologies might impact users' behavior. Even if it might be difficult to apply them in Authors research, they should be at least addressed in the discussion.

Reviewer 3 Report

The paper forwards the idea of applying the AR navigation to the CCI (Culture and Creative Institutions), namely exhibition guidance and analysis. The analysis process is based on the users’ tracks as one of the most valuable evaluation for given hypotheses. However, two hypotheses (Section 4) address generally the same idea (Hyp.2 contradicts Hyp. 1),
so proving/declining one of them gives enough scientific output.

There are a number of already available methods of user’s evaluation for CCI, based usually on  various marketing criteria (number of sold tickets, time spent per display), and technology supporting browsing of collections (i.e. language regionalized audio guides triggered by RFID, QR codes) and the authors refer to game-oriented solutions (Pokemon-Go). The originality of the authors’ approach is in their contribution to enhancing public participation in the outdoor exhibitions (heavy relied on GPS tracking)  with the use of modern technologies like Augmented Reality. AR is characterized by distinctive feature of being ‘immersive’, where user is immersed in the interactions with holograms. On contrary to classic visualizations based on static images and 3d (paper) maquettes, AR gives the feeling of ‘being guided’ through the volume of space and buildings.
Viewing AR simulation may cause potential problems when the system is applied to the meaningless or abstract images but works quite well with images that are recognized and are familiar to the user (humanoid avatar guide). The method proposed by the authors is a development of ‘surrogate’ museum guide maps applied to the displays in the outdoor scenery. Another issue is the lack of interactivity/responsiveness (attaching tags, likes etc to assets/places) in the exhibition landscape. Perhaps that was the reason why there is little difference in guided/non guided  explorations.).
Although the presented approach is illustrated by introductory case-studies it shows a valuable method that can help with design process of outdoor exhibitions.
Addressing some game related  metric would be more very interesting like RV (Replay Value)/ Retention for exhibition and/or  KPI (Key Performance Indicator) that would be useful for creative industries CRM planning...

Round 2

Reviewer 1 Report

Authors adressed the main points of the review. However, the description of the population participating to the study is still missing. Presentation of conclusions could be improved (Section "Discussion").

Reviewer 2 Report

The provided updates are sufficient.